# Plant metabolism of nematode pheromones mediates plant-nematode interactions

Murli Manohar [1], Francisco Tenjo-Castano[1], Shiyan Chen[2], Ying K. Zhang[1,3], Anshu Kumari[1], Valerie M. Williamson[4], Xiaohong Wang[2,5], Daniel F. Klessig [1,2]* & Frank C. Schroeder[1,3]*

Microorganisms and nematodes in the rhizosphere profoundly impact plant health, and small-molecule signaling is presumed to play a central role in plant rhizosphere interactions. However, the nature of the signals and underlying mechanisms are poorly understood. Here we show that the ascaroside ascr#18, a pheromone secreted by plant-parasitic nematodes, is metabolized by plants to generate chemical signals that repel nematodes and reduce infection. Comparative metabolomics of plant tissues and excretions revealed that ascr#18 is converted into shorter side-chained ascarosides that confer repellency. An *Arabidopsis* mutant defective in two peroxisomal acyl-CoA oxidases does not metabolize ascr#18 and does not repel nematodes, indicating that plants, like nematodes, employ conserved peroxisomal β-oxidation to edit ascarosides and change their message. Our results suggest that plant-editing of nematode pheromones serves as a defense mechanism that acts in parallel to conventional pattern-triggered immunity, demonstrating that plants may actively manipulate chemical signaling of soil organisms.

[1] Boyce Thompson Institute, Ithaca, NY 14853, USA. [2] Plant Pathology and Plant-Microbe Biology Section, School of Integrative Plant Science, Cornell University, Ithaca, NY 14853, USA. [3] Department of Chemistry and Chemical Biology, Cornell University, Ithaca, NY 14853, USA. [4] Department of Plant Pathology, University of California, Davis, CA 95616, USA. [5] Robert W. Holley Center for Agriculture and Health, USDA-ARS, Ithaca, NY 14853, USA. *email: dfk8@cornell.edu; schroeder@cornell.edu

mmune responses to pathogen attack in plants and animals are triggered in part by detection of pathogen-associated molecular patterns (PAMPs) by specific extra- and intra-cellular sensors[1–4] (Fig. 1a). This includes evolutionarily conserved macromolecules specific to different classes of pathogens, for example, flagellin and peptidoglycan for bacteria or chitin for fungi[5–7]. Activation of innate immunity resulting from recognition of these foreign (non-self) macromolecules is generally termed pattern-triggered immunity (PTI). In the case of parasitic nematodes, a conserved family of pheromones, the ascarosides, similarly functions as a small-molecule signature that elicits plant immune responses[8].

Nematodes are among the most abundant animals on earth[9], and plant-parasitic nematodes are ubiquitous in soil and parasitize most commercial crops causing annual losses tens of billions of dollars. Ascarosides are glycosides of the dideoxysugar ascarylose that carry a fatty acid-derived lipophilic side chain and are optionally decorated with additional building blocks of diverse metabolic origin[10,11] (Fig. 1b). These glycolipids have been identified almost exclusively from nematodes, including free-living as well as insect, vertebrate, and plant-parasitic nematode species and appear to play a central role in nematode chemical communication, regulating diverse aspects of their development, and behavior[12–19]. Only recently has it become apparent that ascarosides, as an ancient molecular signature of nematodes, are also perceived by other phyla, including fungi and plants[8,20–22]. In plants, exposure of roots or leaf tissue to ascr#18, an ascaroside abundantly produced by several genera of plant-parasitic nematodes, results in rapid activation of the canonical defense signaling pathways, similar to the effects of microbial macromolecular PAMPs[8,23] (Fig. 1a).

While the molecular structures of microbial and nematode-derived PAMPs have been identified, it remains unclear to what extent *editing* of the PAMPs by metabolism in the infected plant or animal plays a role for the observed immune responses (Fig. 1a). For example, plant perception of the 22-amino acid flg22 epitope in bacterial flagellin appears to require removal of its extensive glycosylation coat[24,25], via yet unknown mechanisms. In the case of small-molecule PAMPs, such as ascarosides, uptake of the compounds by the infected plant could result in extensive chemical modification that could be a required part of, or contribute to, the observed defense responses. Uptake of ascr#18 seemed likely, given that activation of PAMP-triggered defense responses in plants usually involves interaction of the PAMP with plasma membrane-localized pattern recognition receptors (PRRs), followed by internalization of the receptor-ligand complex[26,27]. Therefore, we investigated whether plants metabolize nematode-derived ascarosides. Our results show that plants actively partake in nematode chemical communications by biochemical editing of ascarosides and reveals a dual function of these small-molecule PAMPs in plant defense.

## Results

**Plants metabolize ascr#18 into a blend of ascarosides**. To test whether plants metabolize the nematode-derived PAMP ascr#18, we employed comparative metabolomics based on high-resolution liquid chromatography–mass spectrometry (LC-MS) analyses of ascr#18-treated plant tissues. Since plants would naturally encounter nematodes via their roots, we used dicotyledon tomato and Arabidopsis and monocotyledon wheat plants grown under sterile conditions. Plant roots were soaked in aqueous ≤ 0.1% ethanol-containing solution without (mock) or with ascr#18 for 24 h. Subsequently, root tissue was harvested, extracted, and analyzed by LC-MS. To compare datasets from mock- and ascr#18-treated plants, we employed the XCMS

comparative metabolomics software package[28], focusing on peaks that were entirely absent in mock-treated plants and thus could represent ascr#18-derived metabolites. The software-generated lists of differential features were then manually curated to remove false positives, as well as isotope peaks and mass spectrometric adducts. For all three plant species, this analysis revealed the presence of ascr#18 in ascr#18-treated but not in mock-treated plants (Fig. 1c, d). In addition, in all three species, we found a series of additional peaks that were present only in ascr#18-treated plants (Fig. 1c, d and Supplementary Fig. 1). By comparing molecular ion weights, mass spectrometry (MS)/MS spectra, and retention times with those of known ascarosides[10,29], we determined that these additional peaks represent ascarosides with shorter side chains, specifically ascr#10, ascr#1, and ascr#9 (Supplementary Figs. 2 and 3). In all three plant species, the distribution of ascr#18 metabolites was similar at all tested ascr#18 concentrations, including low nanomolar concentrations, which we had previously shown to fall in the physiological range[8,23]. The ascaroside ascr#9, in which the 11-carbon side chain of ascr#18 had been shortened to only 5 carbons, was the most abundant metabolite in all cases (Fig. 1d and Supplementary Fig. 1). To corroborate that the shorter-chained ascarosides are in fact derived from the added ascr#18, we repeated the experiment in tomato using $^{13}C_2$-labeled ascr#18 (Fig. 2a). Tomato plants root-treated with $^{13}C_2$-ascr#18 produced $^{13}C_2$-labeled ascr#9, confirming that the shorter-chained ascarosides are derived from the added ascr#18. Whereas the initial experiments were conducted using sterile growth media, both Arabidopsis and tomato metabolized ascr#18 to ascr#9 similarly in a field soil/potting soil mix (Supplementary Fig. 4a, b). Moreover, ascr#18 was metabolized into ascr#9 by tomato roots naturally infested with root-knot nematodes (Supplementary Fig. 4c).

Although plants grown in sterile media using surface-sterilized seeds consistently metabolized ascr#18 to ascr#9, we considered the possibility that microorganisms, e.g., endophytic microbes associated with the seeds or roots, could play a role. To test whether soil-associated microbes are required for the observed metabolic transformations, we directly infiltrated leaves of 4-week-old soil grown Arabidopsis plants with ascr#18 and subsequently harvested leaf tissue for analysis by LC-MS as above. Similar to root treatment, ascr#18 accumulated and was converted into shorter side-chained ascarosides in the leaves (Fig. 2b, c), About 50% of ascr#18 was metabolized during the first 12 h, during which time we observed concomitant accumulation of ascr#9. Formation of ascr#9 was also observed in leaves of 4-week-old tomato plants infiltrated with ascr#18 (Supplementary Fig. 5). Measured ascr#9 concentrations peaked during the first 12 h post infiltration in Arabidopsis and subsequently declined to very low levels at 96 h, indicating that ascr#9 was metabolized further or transported to other tissues in plants (Fig. 2c). Taken together, these experiments demonstrate uptake of ascr#18 and conversion into ascarosides with a shorter side chain, predominately ascr#9, in both monocots and dicots.

**Plants metabolize ascr#18 via peroxisomal β-oxidation**. Although metabolism of ascr#18 into ascr#9 was observed in several different plant species and under different conditions, involvement of plant-associated endophytes cannot be entirely excluded based on the experiments described above. To obtain unambiguous proof for the participation of the plant in ascr#18 metabolism, we thus pursued identification of the putatively involved plant enzymes. Inspection of the structures of the identified ascr#18-derived metabolites revealed that their side chains are two, four, and six carbons shorter than the side chain of ascr#18 (Fig. 1c). This observation suggested that these

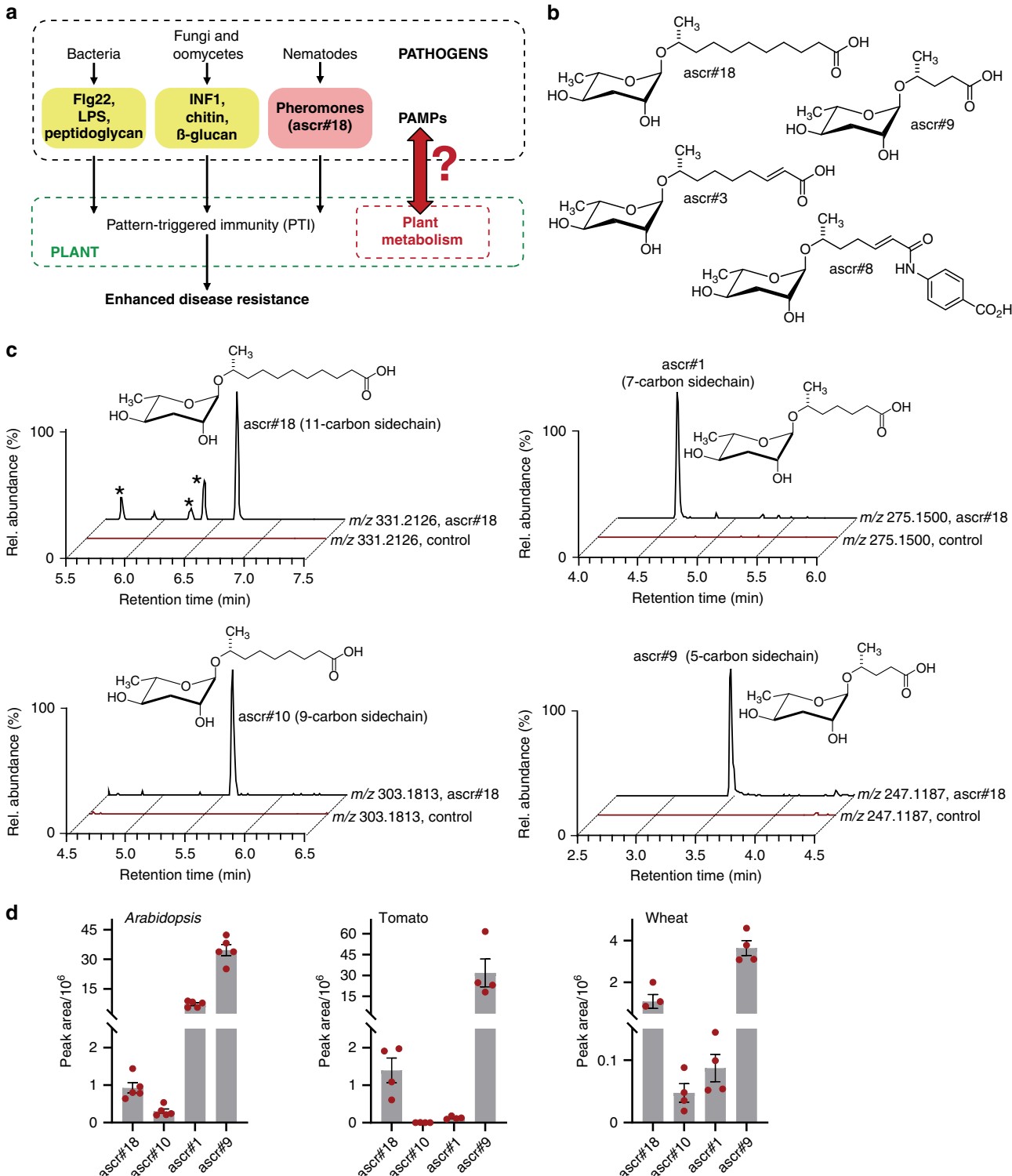

**Fig. 1 Accumulation and metabolism of nematode-derived ascr#18 in plants. a** Recognition of nematode-derived ascr#18 activates pattern-triggered immunity (PTI) in plants, similar to pathogen-associated molecular patterns (PAMPs) derived from other microbes. **b** Examples for ascarosides previously discovered in *C. elegans* and other nematode species. **c** LC-MS analysis of *Arabidopsis* roots treated with 1 μM ascr#18 for 24 h, showing extracted ion chromatograms [EIC] in ESI⁻ of ascr#18, ascr#10, ascr#1, and ascr#9. Peaks marked with an asterisk represent unrelated metabolites of similar *m/z*. **d** Accumulation of ascarosides in *Arabidopsis*, tomato, and wheat roots treated with 1 μM ascr#18 for 24 h. Abundances of ascarosides are shown as the peak area, as measured in LC-MS. Data are mean ± SEM ($n = 5$) (also see Supplementary Fig. 1). Source data are provided as a source data file.

| compounds may be derived from ascr#18 via β-oxidation. Mitochondrial and peroxisomal β-oxidation are highly conserved metabolic pathways that iteratively shorten straight-chain fatty acids in two-carbon increments. In nematodes, ascarosides, | including ascr#18 and the identified ascr#18 metabolites, are produced via peroxisomal β-oxidation from longer-chained precursors[29]. Given the nematode precedent, and since fatty acid degradation in plants mostly occurs via the β-oxidation pathway |

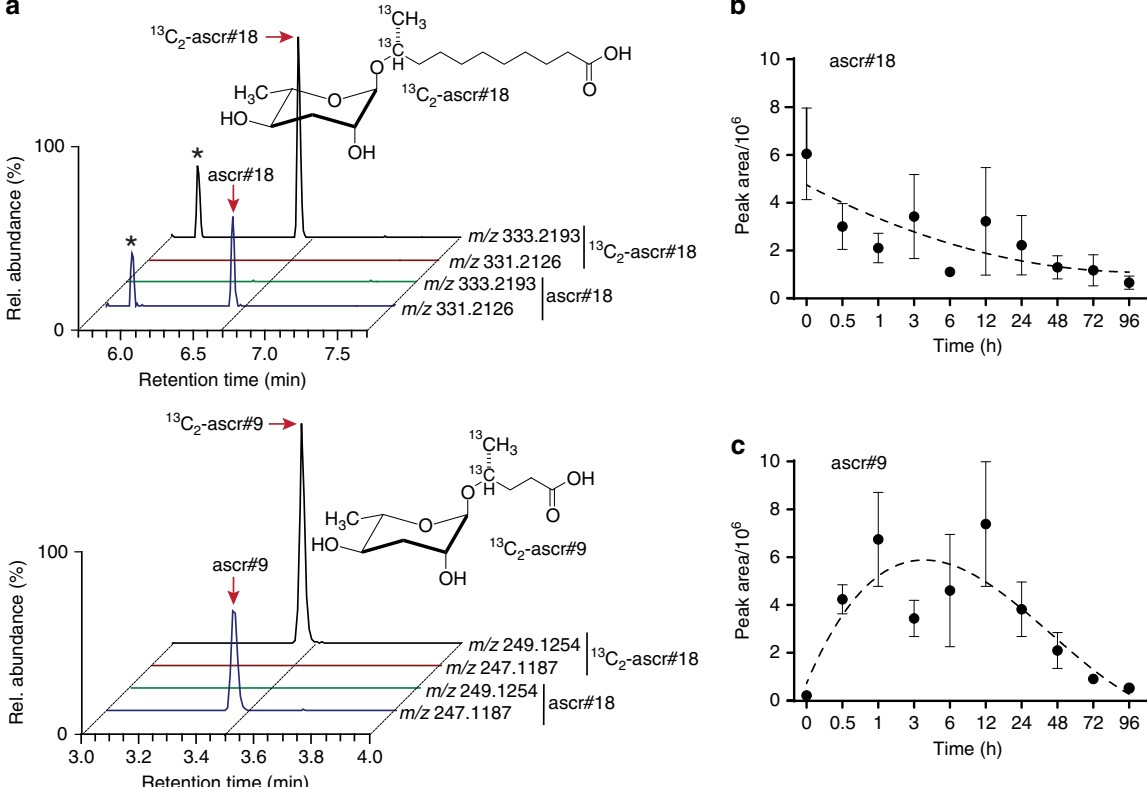

**Fig. 2 Short-chained ascarosides are derived from the metabolism of exogenously applied ascr#18. a** LC-MS analysis of tomato roots treated with ascr#18 or $^{13}C_2$-labeled ascr#18. Plants treated with $^{13}C_2$-labeled ascr#18 show peaks at 333.21932 and 249.12542, representing [M-H]$^-$ of $^{13}C_2$-ascr#18 and $^{13}C_2$-ascr#9, but not at 331.21261 and 247.11871, which correspond to [M-H]$^-$ of ascr#18 and ascr#9, and vice versa. Structures of ascr#18 and ascr#9 indicating the positions of the $^{13}C_2$-label are shown above ion chromatograms. Peaks marked with an asterisk represent unrelated peaks. **b, c** Relative abundances of ascr#18 and ascr#9 as measured by LC-MS. Four-week-old *Arabidopsis* leaves were infiltrated with 1 µM ascr#18, and leaf tissue were harvested for LC-MS analyses over a period of 96 h post infiltration of ascr#18. Data are mean ± SEM ($n = 3$) (also see Supplementary Fig. 4). Source data are provided as a source data file.

in the peroxisomes[30], we hypothesized that metabolism of ascr#18 in plants may also proceed via peroxisomal β-oxidation. Plant peroxisomal β-oxidation has been extensively characterized genetically and plays an important role in the biosynthesis of signaling molecules. For example, peroxisomal β-oxidation contributes to the biosynthesis of jasmonic acid and auxin[31]. A general scheme comparing the peroxisomal β-oxidation pathways in *Arabidopsis* and *C. elegans* is shown in Fig. 3a.

To test whether ascr#18 is metabolized via peroxisomal β-oxidation in plants, we analyzed *Arabidopsis* mutants impaired in key enzymes of this pathway (Fig. 3b). We found that metabolism of ascr#18 into shorter-chained ascarosides was dramatically reduced in an *Arabidopsis* mutant defective in two of the six annotated acyl-CoA oxidases, *ACX1* and *ACX5*[32]. Partial abolishment of *ACX1* and *ACX5* transcription in the *acx1 acx5* mutant had been previously demonstrated[32] and was confirmed by RNA-Seq analysis (Supplementary Fig. 6). In contrast, two other putative peroxisomal β-oxidation genes, *ibr10* and *ech2*, a putative enoyl-CoA hydratases[31–33], are not required for ascr#18 metabolism in *Arabidopsis*. ACXs participate in the second step of peroxisomal β-oxidation by introducing α,β-unsaturation in the side chain[29,31]. Detailed metabolomic comparison of ascr#18-treated wildtype and *acx1 acx5* plants revealed that the mutant accumulated elevated levels of ascr#18 (11-carbon side chain), as well as smaller amounts of ascr#10 (9-carbon side chain), whereas ascr#1 and ascr#9 (7- and 5-carbon chains, respectively) were not detected (Fig. 3c). To test whether ACX1 and/or ACX5 are required for ascr#18 metabolism, we also analyzed ascr#18-

treated *acx1* and *acx5* single mutants and found that in both single mutants ascr#18 metabolism proceeds largely unimpaired, resulting in accumulation of ascr#9 very similar to the wildtype control (Supplementary Fig. 7). This result is in line with previous reports that enzymes associated with peroxisomal β-oxidation, including acyl-CoA oxidases, often function redundantly[30,32]. Taken together, these observations indicate that ascr#18 metabolism in plants into shorter-chained ascarosides proceeds via endogenous peroxisomal β-oxidation, and that ACX1 and ACX5 are functionally redundant for chain shortening of ascr#10, whereas the first chain-shortening step from ascr#18 to ascr#10 may involve additional acyl-CoA oxidases.

**Ascr#18 metabolism is required for nematode resistance.** Given the rapid conversion of ascr#18 into shorter-chained ascarosides via peroxisomal β-oxidation, we asked whether metabolism of ascr#18 is required for the activation of defense responses and resistance to pathogens. In previous work, we showed that ascr#18 treatment enhances resistance of monocot and dicot plants to a wide range of pathogens, including bacteria, viruses, fungi, oomycetes, and nematodes[8]. To test whether ascaroside metabolism plays a role in ascr#18-mediated enhanced resistance to a bacterial pathogen, we compared the effect of ascr#18 treatment on infection with *Pseudomonas syringae* DC3000 in wildtype *Arabidopsis* and the *acx1 acx5* mutant, which is defective in metabolism of ascr#18 to ascr#9. Pretreatment with 1 µM ascr#18 for 24 h prior to infection with *P. syringae* provided comparable levels of protection in *acx1 acx5* and wildtype

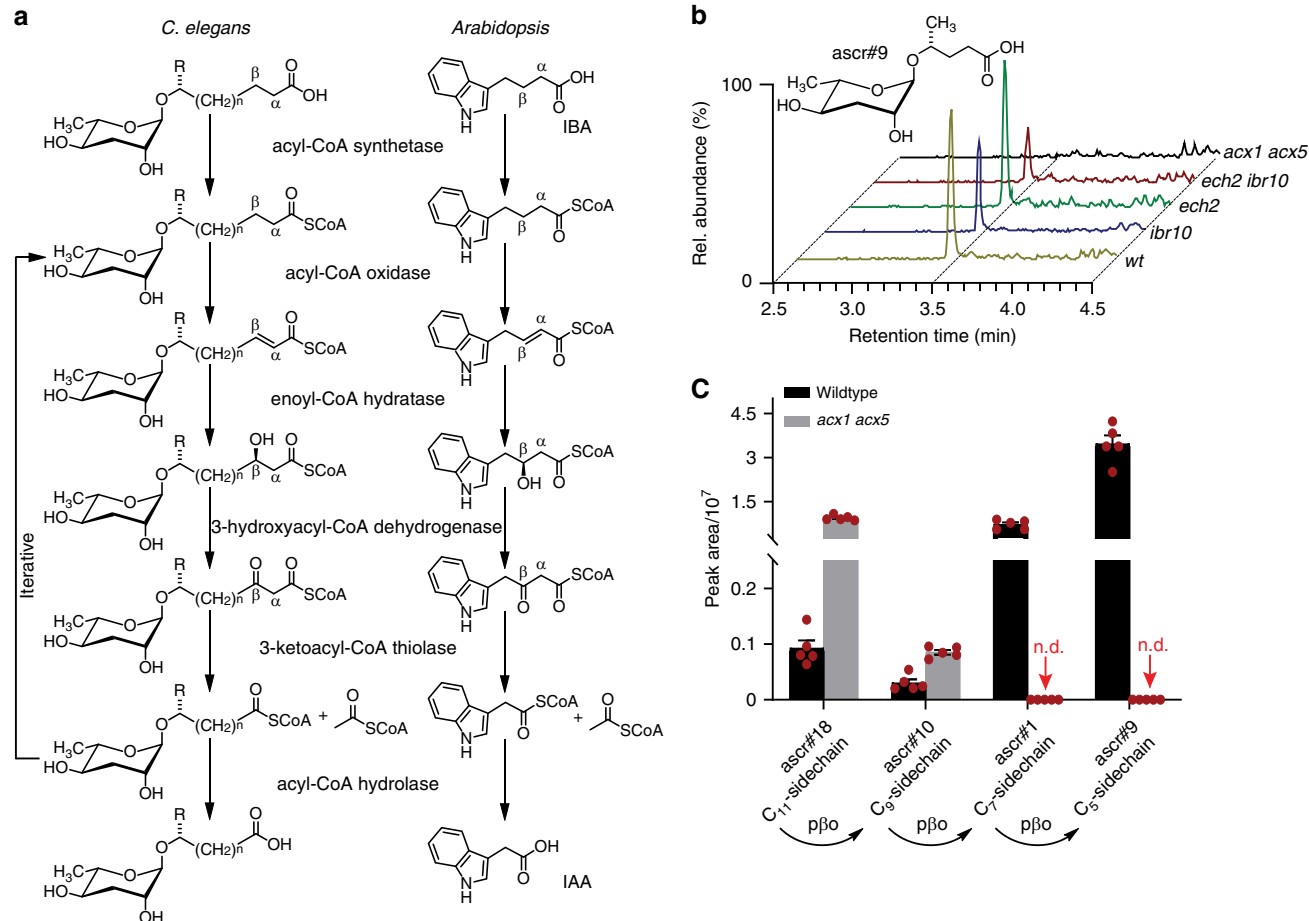

**Fig. 3 *Arabidopsis* metabolizes ascr#18 via conserved peroxisomal β-oxidation. a** Comparison of peroxisomal β-oxidation pathways and associated enzymes involved in the conversion of indole-3-butyric acid (IBA)-to-indole acetic acid (IAA) in *Arabidopsis* and ascaroside biosynthesis in *C. elegans*. **b** LC-MS analysis of *Arabidopsis* seedlings treated with 1 μM ascr#18 for 24 h before sample extraction, comparing ascr#9 production in wildtype and *ibr10, ech2, ech2 ibr10*, or *acx1 acx5* mutants. **c** Comparison of ascaroside abundances in *Arabidopsis* wildtype and *acx1 acx5* roots treated with 1 μM ascr#18. Data are mean ± SEM ($n = 5$); n.d. = not detected. Source data are provided as a supplemental data file. Source data are provided as a source data file.

(Fig. 4a, b), indicating that metabolism of ascr#18 via *ACX1* or *ACX5* is not required for enhanced resistance against this bacterial pathogen.

To assess whether plant metabolism of ascr#18 contributes to defense against nematodes, we compared the effect of ascr#18 treatment on infection of wildtype and the *acx1 acx5* mutant with root-knot nematode (*Meloidogyne incognita*). We found that, whereas pretreatment of roots with 10 nM or 50 nM ascr#18 for 48 h prior to inoculation provided significant protection in wildtype, ascr#18 treatment had no effect in the *acx1 acx5* mutant (Fig. 4a, c).

In a second experiment, wildtype and *acx1 acx5* were treated with ascr#18 for 48 h prior to moving seedlings to a PF-127 gel containing ~200 *M. incognita* second-stage larvae (J2) (Fig. 4a, d, e). J2 larvae touching the roots were counted at 6 h post seedling transfer. In response to pretreatment with ascr#18 at 50 nM, numbers of J2 larvae touching the root tips or whole root area were significantly reduced in wildtype, but not in *acx1 acx5*. A second, much higher ascr#18 concentration (1 μM) did not significantly affect *M. incognita* behavior in either wildtype or mutant. Taken together, these results indicate that ascr#18 metabolism is required for the enhanced resistance of ascr#18-treated plants to nematode infection, whereas enhanced resistance to bacteria is not affected.

**Defense signaling is independent of ascr#18 metabolism.** Next, we aimed to clarify whether differences in activation of defense signaling in wildtype and *acx1 acx5* plants underlie the observed differences in resistance to nematodes. In previous work, we had shown that in *Arabidopsis*, treatment with low micromolar ascr#18 concentrations results in activation of conserved defense signaling pathways in monocots and dicots, including mitogen-activated protein kinase (MAPK) signaling and the jasmonic acid (JA), and salicylic acid (SA) signaling pathways[8]. To assess whether ascr#18 metabolism via peroxisomal β-oxidation is required for activation of these defense signaling pathways, we compared expression of selected marker genes in leaves of ascr#18-treated wildtype and *acx1 acx5* plants (Fig. 4f and Supplementary Table 1). Treatment with ascr#18 induced components of JA signaling (*Plant Defensin1.2 (PDF1.2), Allene Oxidase Synthase (AOS)*, and *Lipooxygenase2 (LOX2))* and SA signaling ((*Pathogenesis-Related1 (PR-1)* and *Pathogenesis-Related4 (PR-4))*, and *WRKY transcription factor 53 (WRKY53))* to similar extents in wildtype and *acx1 acx5*, although induction of the MAPK-related *Flg22-Induced Receptor Kinase1 (FRK1)* was slightly weaker in the mutant. These findings indicate that ascr#18 metabolism via peroxisomal β-oxidation is generally not required for activation of canonical defense signaling pathways by ascr#18, and our observation that enhanced protection of *Arabidopsis*

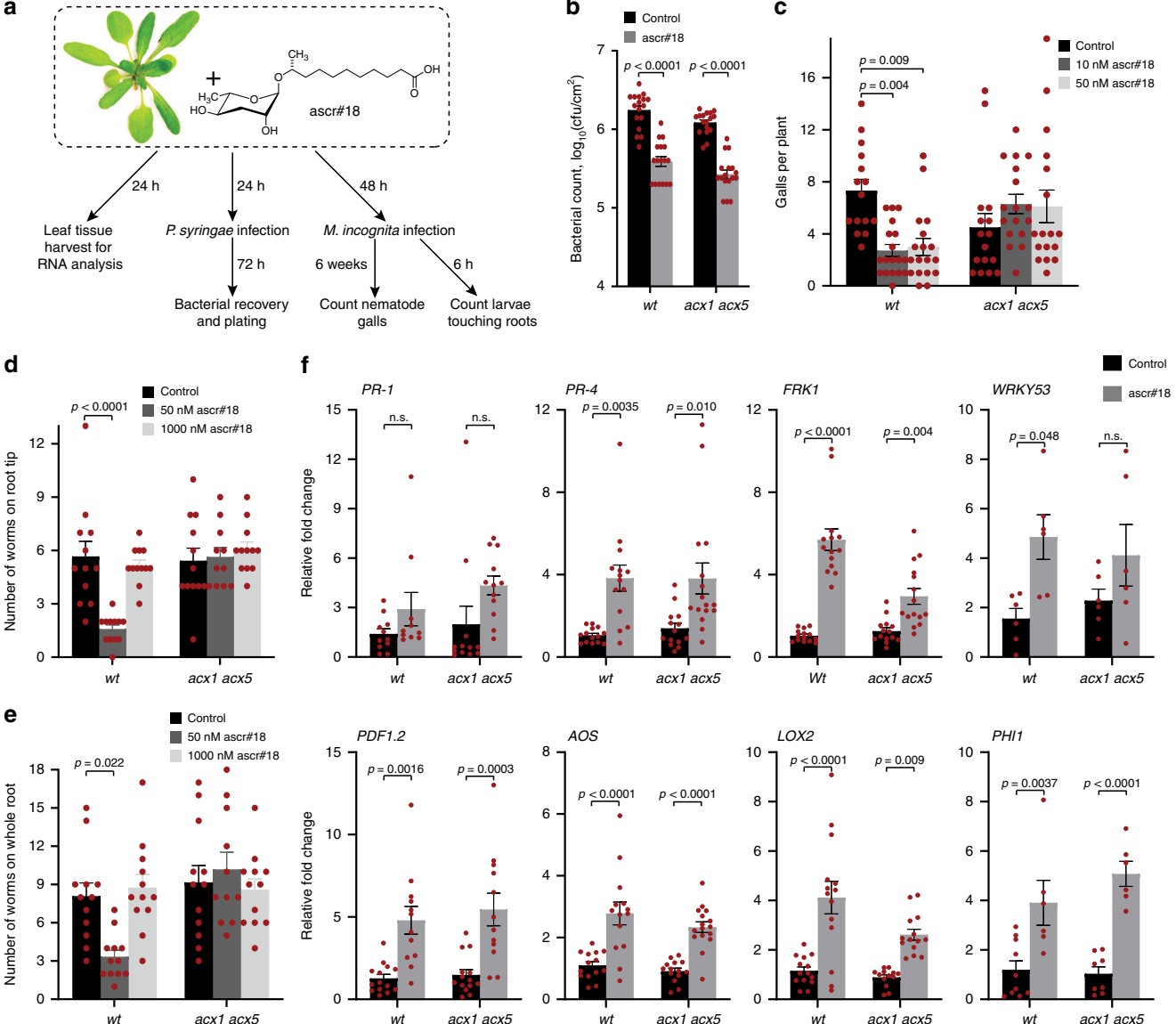

**Fig. 4 acx1 acx5 is required for ascr#18-mediated enhanced resistance to root-knot nematodes but not a bacterial pathogen. a** Experimental designs for assessing activation of defense pathways and resistance to nematodes and bacteria. **b** Enhanced resistance to virulent *P. syringae pv. tomato (Pst)* DC3000 does not require *acx1 acx5*. Bacterial growth was assayed 3 days post inoculation. Data are averages ± SEM (*n* = 17). **c** ascr#18 increases resistance of *Arabidopsis* wildtype, but not *acx1 acx5* mutants, to plant-parasitic nematodes (*M. incognita*). *Arabidopsis* seedlings were treated with buffer or the indicated concentrations of ascr#18 for 48 h before inoculation with ~300 freshly hatched *M. incognita* J2 larvae. The numbers of root galls of infected plants were counted 6 weeks post inoculation. Data are averages ± SEM (*n* ≥ 15). **d, e** ascr#18 treatment of *Arabidopsis* wildtype, but not *acx1 acx5* mutants results in deterrence of *M. incognita* J2 larvae resistance. *Arabidopsis* seedlings were treated with the indicated concentrations of ascr#18 for 48 h before transfer into 12-well plates containing Pluronic F-127 gel with ~200 freshly hatched *M. incognita* J2 larvae. Larvae touching root tips (**d**) or the whole area of roots (**e**) were counted at 6 h post seedling transfer. Data are average ± SEM (*n* = 12). **f** Induction of defense–response genes in *Arabidopsis* leaves 24 h after root treatment with 1000 nM ascr#18. Transcript levels were determined by qRT-PCR. Data are mean ± SEM (*n* ≥ 6). Adjusted *p*-values were calculated by two-way ANOVA followed by Tukey multiple comparisons post hoc test. n.s. = not significant. Source data are provided as a source data file.

against *P. syringae* is independent of ascr#18 metabolism is consistent with this result. Consistent with a previous study[8], expression of defense genes was not induced by low (nanomolar) ascr#18 concentrations in wildtype and remained unchanged in the *acx1 acx5* double mutant (NCBI−SRA RNA-Seq submission: PRJNA550121, Supplementary Fig. 8). Given that enhanced resistance against nematodes is observed at low ascr#18 concentrations that are insufficient to induce defense gene expression in wildtype or the *acx1 acx5* double mutant, it appears that the effects of ascr#18 metabolism on nematode resistance may not be directly mediated by plant defense signaling pathways.

**Plant roots secrete ascr#18 metabolites.** Plants secrete a large array of primary and secondary metabolites via their roots, including signaling molecules that facilitate interactions with soil biota[34]. Since ascr#18 metabolism is not required for resistance against bacteria or activation of the canonical plant defense signaling pathways, but nonetheless is essential for defense against nematodes, we considered the possibility that ascr#18-derived metabolites are secreted via the roots and thereby affect nematode host-finding behavior.

To test this possibility, we compared root-secreted metabolites of ascr#18- and mock-treated *Arabidopsis* (Fig. 5a). Ten-day-old

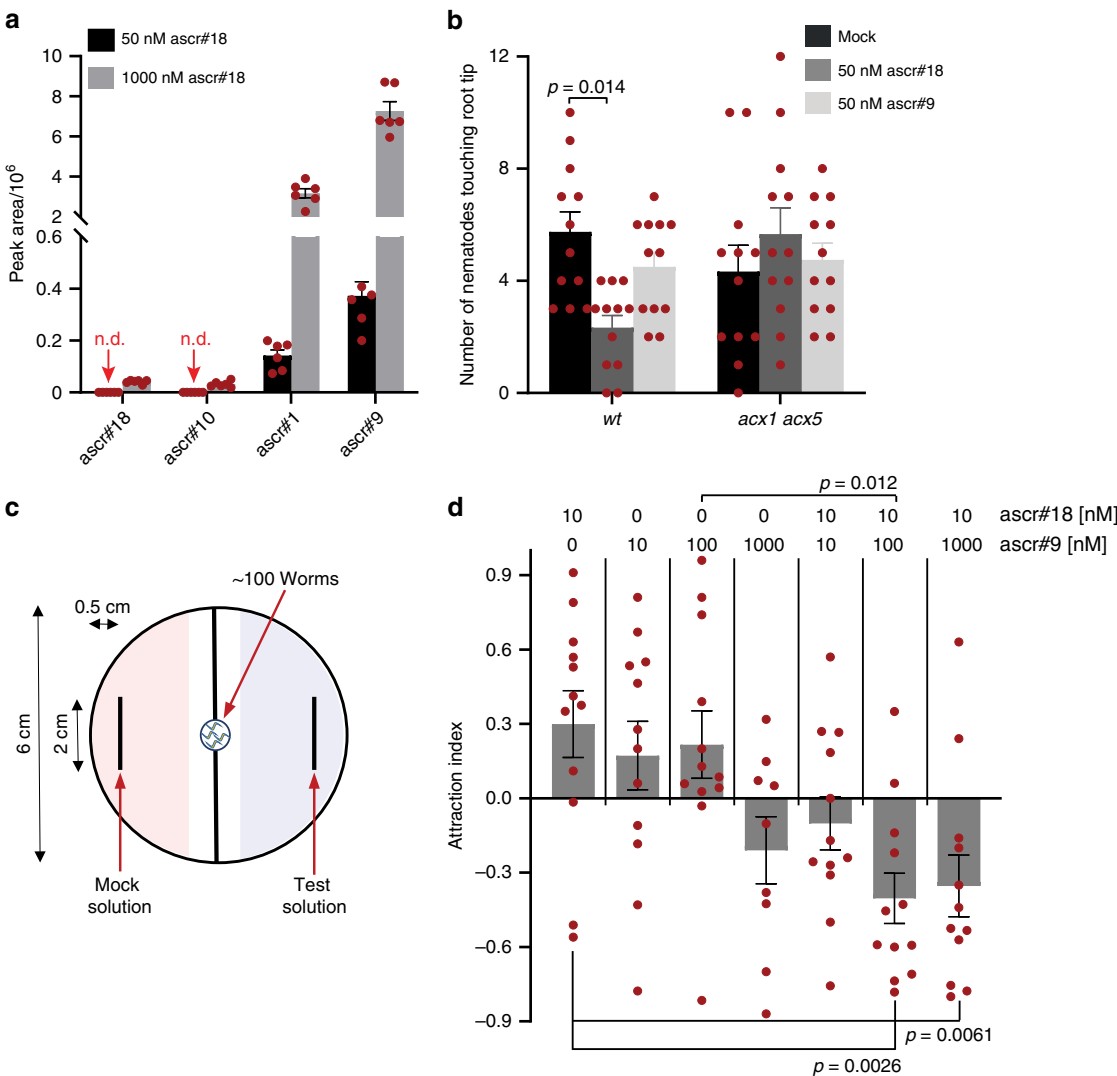

**Fig. 5 Plant-derived ascaroside blends deter root-knot nematodes. a** Relative abundances of ascarosides in root exudates of *Arabidopsis* treated with 50 and 1000 nM ascr#18, as determined LC-MS. Data are average ± SEM (*n* = 6), n.d. = not detected. **b** *Arabidopsis* wildtype and *acx1 acx5* were treated with 50 nM ascr#18 or ascr#9 for 48 h before transfer into 12-well plates containing Pluronic F-127 gel with ~200 freshly hatched *M. incognita* J2 larvae. Larvae touching the terminal part of roots were counted at 6 h after seedling transfer. Data are average ± s.d. (*n* = 12). **c** A small volume (10 μL) of ascr#18, ascr#9, or ascr#18/ascr#9 solutions was placed on one side of a 6-cm Petri dish with mock solutions on the other side. Subsequently, ~100 freshly hatched *M. incognita* J2 larvae were placed in the center of the plate. Larvae in the indicated scoring areas were counted after 4 h. **d** Attraction index for different concentrations of ascr#18, ascr#9, or ascr#18/ascr#9 mixtures measured using the layout shown in **c**. Data are mean ± SEM (*n* = 12). Adjusted *p*-values in Fig. 5b were calculated by two-way ANOVA followed by Tukey multiple comparisons post hoc test. For Fig. 5d, adjusted *p*-values were calculated by one-way ANOVA followed by Tukey multiple comparisons post hoc test. Source data are provided as a data file.

*Arabidopsis* seedlings were treated in growth media supplemented with ascr#18 for 6 h prior to collection of exudates from roots submerged in water. HPLC-MS-based comparative metabolomic analyses revealed secretion of all of the identified ascr#18 metabolites, including ascr#10, ascr#1, and ascr#9, in ascr#18-treated wildtype. Consistent with the above results, only ascr#10 was detected in *acx1 acx5* in addition to larger amounts of residual ascr#18 than in wildtype (Supplementary Fig. 9a). Analogous results were obtained in experiments with tomato roots (Supplementary Fig. 9b, c). High-resolution liquid chromatograph mass spectrometer (HRLC-MS) analysis of growth media showed a steady buildup of short-chained ascarosides over a period of 48 h post treatment, indicating constant uptake, conversion, and excretion of ascarosides through the root (Supplementary Fig. 9d). In both tomato and *Arabidopsis*, ascr#9 was the most abundant root-excreted ascr#18 metabolite, whereas ascr#10 was least abundant.

**Plant-derived ascaroside blends deter parasitic nematodes.** Several previous studies have demonstrated that distinct blends of ascarosides can induce attraction or avoidance behaviors. For example, the most abundant ascr#18 metabolite, ascr#9, has been previously shown to mediate dispersal behavior in entomo-pathogenic nematode species[19], which are phylogenetically related to *Meloidogyne* spp[9,35,36]. Therefore, we suspected that the secretion of ascr#9 may play a role in mediating plant–nematode interactions. However, in contrast to treatment of wildtype with ascr#18, treatment of neither wildtype nor *acx1 acx5* with ascr#9 for 48 h before moving seedlings to PF-127 gel containing *M. incognita* J2 larvae significantly reduced infection (Fig. 5b). Since several previous studies have shown that ascaroside-mediated phenotypes may involve synergism of two or more components[14,19,37], we next considered the possibility that a blend of ascarosides, not just one compound, is responsible for

the observed suppression of nematode migration towards roots, Therefore, we tested a series of combinations of ascr#18, which is directly excreted by the nematodes, and its most abundant plant metabolite, ascr#9, in a previously described quantitative chemotaxis population assay[38] (Fig. 5c). We found that blends containing a 1:10 ratio of ascr#18 to ascr#9 elicited significant avoidance behavior, whereas low concentrations of either compound are slightly attractive or have no effect (Fig. 5d). These results indicate that secretion of ascr#9 by the plant in combination with ascr#18 repels nematodes in the rhizosphere and thereby reduces infection.

## Discussion

Soil micro- and macrobiota form complex communities that rely on extensive chemical communication networks[34,39]. Their composition profoundly impacts plant development and health, and plants have evolved diverse strategies to interact with beneficial microorganisms and importantly, to recognize and protect against pathogens[40]. In turn, microbial pathogens have evolved diverse mechanisms to counteract plant defense systems, in what has been referred to as an evolutionary "arms race" between plants and their pathogens[1]. Early perception of pathogen-derived PAMPs by cell-surface-localized PRRs is a critical early step in plant defense. Recently, we demonstrated that plants detect ascr#18, the major ascaroside excreted by plant-parasitic nematodes, as a nematode-derived PAMP that elicits strong immune responses and provides protection against a wide range of pathogens, including parasitic nematodes (Fig. 1a)[8,23]. Similar to microbe-derived flagellin or lipopolysaccharide, ascr#18 elicits plant defense responses in monocots and dicots at very low concentrations. However, in contrast to these macromolecular microbial PAMPs, ascarosides are small molecules that represent a highly conserved class of nematode pheromones.

Here, we uncover another dimension in the plant response to PAMPs (Fig. 6). We show that both monocot and dicot plants convert ascr#18 rapidly into shorter side-chained ascarosides that act as chemical signals when excreted into the rhizosphere to regulate early stages of plant–nematode interactions. These data highlight an evolutionary conserved chemotropic mechanism that regulates the pre-invasion phase of plant–nematode interactions. Parallel activation of PTI and generation of repellent signals indicates a two-pronged strategy to cope with pathogen infection. In turn, avoidance behavior of nematodes to mixtures of ascr#18 and plant-derived ascr#9 may be an important component of nematode host-finding behavior, preventing overpopulation of already infected plants. In that sense, invading nematodes may highjack plant peroxisomal β-oxidation to strategically limit further infection.

Activation of PTI by ascr#18 likely involves interaction with specific PRRs. PTI activation by microbial PAMPs requires clathrin-dependent endocytosis of PAMP-PRR complex, which is then transported to vacuoles[26]. Internalization of ascr#18 may follow a similar route; however, additional steps may be required to transport the putative ascr#18-PRR complex to the peroxisome for subsequent processing via β-oxidation.

Conversion of ascr#18 into shorter side-chained ascarosides provides the first example in which enzymatic editing of a PAMP molecule *in planta* is required for PTI. We demonstrate that editing of ascr#18 proceeds via peroxisomal β-oxidation in *Arabidopsis* and that a mutant defective in two peroxisomal acyl-CoA oxidases is defective in ascr#18-triggered defense against nematodes. Metabolism of ascr#18 in plants is reminiscent of ascaroside biosynthesis via peroxisomal β-oxidation in *C. elegans*[29]. Peroxisomal β-oxidation is highly conserved in animals and plants and plays a central role in energy metabolism, as well as diverse signaling pathways, e.g., by contributing to the

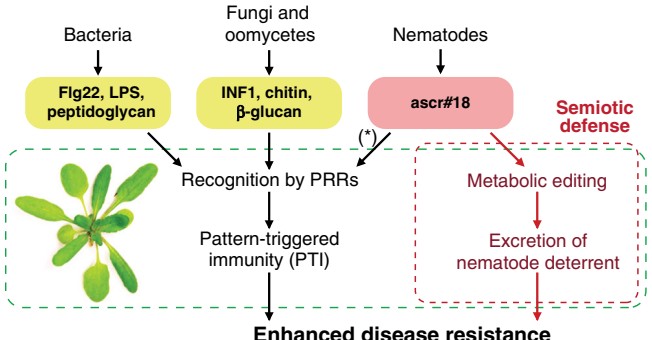

**Fig. 6 Editing of ascr#18 and generation of repellent signals act in parallel with conventional innate immune responses in plants.** Plants detect PAMPs such as flg22, lipopolysaccharide (LPS), and peptidoglycan derived from bacteria, infestin 1 (INF1), chitin, and β-glucan derived from fungi/oomycete, and ascr#18 derived from nematodes via cell-surface-localized PRRs to induce conventional PTI. A parallel "semiotic defense" depends on metabolic editing of nematode-derived ascr#18 by the plant to generate a cocktail of ascarosides pheromones that acts as a repellent of parasitic nematodes (*PRRs unknown).

biosynthesis of the plant hormones auxin and JA[30–32]. Analogous to peroxisomal β-oxidation in *C. elegans* and other animals, acyl-CoA oxidases (ACX) catalyze the first step of the plant peroxisomal β-oxidation cycle and essentially determine the flux of metabolites through this pathway, although biochemical characterization of peroxisomal β-oxidation in *Arabidopsis* remains incomplete[30–33,41,42]. The *Arabidopsis* genome contains six *ACX* paralogs, of which four (*ACX1-4*) have been characterized in greater detail. *ACX1* has medium-to-long chain substrate specificity, with *ACX5* sharing nearly 85% of sequence identity. Thus, they are presumed to be functionally similar and likely are the result of gene duplication[30,32]. Our finding that ascr#18 metabolism is wildtype-like in both *acx1* and *acx5*, but significantly defective in *acx1 acx5* indicates functional redundancy of these two acyl-CoA oxidases. Previous work demonstrated reduced suppression of root elongation in response to IBA in *acx1*, suggesting a disruption of the IBA β-oxidation pathway, whereas the *acx5* single mutant responds similarly to wildtype. However, IBA responses were further reduced in the *acx1 acx5* double mutant compared to *acx1*, indicating that ACX5 contributes to IBA β-oxidation in the absence of ACX1[32], also suggesting partial functional redundancy.

Perhaps as a result of altered auxin and/or jasmonate signaling, the *acx1 acx5* double mutant is more susceptible to insect damage, whereas susceptibility to fungal infection is not significantly different from wildtype[43,44]. Although, the canonical plant defense signaling pathways are not perturbed by low concentrations of ascr#18 in wildtype or *acx1 acx5* and thus do not appear to be directly involved in ascr#18-mediated enhanced nematode resistance, altered auxin or jasmonate signaling could affect overall susceptibility of *acx1 acx5* to nematode infection, which appears to be slightly lower than wildtype (Fig. 4c). Notably, resistance to bacterial pathogens in response to higher concentrations of ascr#18, which significantly increases defense gene expression, remains unaffected in the *acx1 acx5* double mutant.

Rapid metabolism of ascr#18 by plants suggests that evolution of plant peroxisomal β-oxidation may have been shaped in part by selective advantages conferred by the capability to interfere with nematode chemical communication. In this study, we showed that ascr#18 in combination with its most abundant plant metabolite, ascr#9, deters plant-parasitic nematodes. However, it is possible that intermediates in the pathway from ascr#18 to

ascr#9 also contribute, e.g., ascr#1, which is excreted alongside ascr#9 in root exudates.

Enzymatic editing by infected plants or animals likely also plays a role in the perception of microbial PAMPs. For example, plant perception of bacterial flagellin via the conserved flg22 epitope is presumed to involve extensive deglycosylation[24,25], although the enzymes involved have not been identified[26,27]. Protection against cutworms by a glycoside derived from volatile (Z)-3-hexanol emitted by neighboring damaged plants[45] is another example, suggesting that plants may utilize a broad spectrum of metabolic transformations to edit foreign signals for their protection. Our study provides a first glimpse into the functions that metabolism of PAMPs following uptake by the plant may serve in plant–pathogen interactions. Given the pervasiveness of chemical communication networks in the soil, it seems likely that plants not only "listen in" on chemical signals produced by soil micro- and macrobiota, but also actively partake in the semiotic dialog via metabolic editing.

## Methods

**Plant material and growth conditions**. Unless otherwise stated, *Arabidopsis thaliana* ecotype Col-0, tomato (*Solanum lycopersicum*) cultivar M82, and wheat (*Triticum aestivum*) cultivar Kanzler plants were grown in a growth chamber under a 16-h light/8-h dark (22 °C) regime with 70% relative humidity. For sterile growth conditions, seeds were surface sterilized by soaking with a 50% bleach solution for 5–10 min and washed extensively with sterilized water before planting into the plant growth media. *Arabidopsis* growth media contained 2.15 g/L Murashige & Skoog salts (Sigma-Aldrich), 10 g/L sucrose, and the pH was adjusted to 6.0 using KOH. Tomato and wheat growth media contained 4.3 g/L Murashige & Skoog salts, 30 g/L sucrose, 0.112 g/L Gamborg's B5 vitamin solution (Sigma-Aldrich), and the pH was adjusted to 5.5 using KOH. For solid growth media, 8 g/L agar (Sigma-Aldrich) was added before autoclaving.

**Plant genotypes**. *Arabidopsis thaliana* mutant genotypes in the Col-0 background, ibr10-1, ech2-1, ech2-1 ibr10-1, and acx1-2 acx5-1, were obtained from Prof. Bonnie Bartel at Rice University, Houston, Texas, USA[31,32] and homozygous acx1 (CS66497) and acx5 (CS66498) genotypes were obtained from *Arabidopsis* Biological Resource Center (Ohio State University) and demonstrated in Schilmiller et al.[43].

**Plant extraction**. Plant tissues were frozen in liquid nitrogen immediately following collection, ground to a fine powder, and extracted with 350 μL of a mixture of water/methanol/chloroform (1:2:1) for 12 h at 4 °C, with shaking at 220 rpm. Extracts were then concentrated in a speed vacuum concentrator and reconstituted in 100 μL of methanol for LC-MS analysis. For root tissues, seedlings were extensively washed with excess distilled water and gently dried on filter paper. Root tissues from 40 seedlings were pooled in one tube and then extracted as above. For collection of root exudates, seedlings were treated by supplementing ascr#18 into the growth media for 6 h and washed thereafter with water before placing the roots into distilled water for root exudate collection for 1 h. Exudates from ~40 ten-day-old *Arabidopsis* or 40 eight-day-old tomato seedlings were pooled in one tube. The collected root exudate was then freeze-dried and extracted as described above. For tomato plants infested with root-knot nematodes, loose soil was carefully removed and exudate was collected by washing roots with methanol. The methanol containing root exudate was then dried in a rotary evaporator and reconstituted in 250 μL of methanol for LC-MS analysis.

**C. elegans extracts**. Mixed stage *C. elegans* cultures were grown in 25 mL of S-complete medium with *E. coli* strain OP50 for 3 days shaking at 22 °C and 220 rpm. Subsequently, the cultures were centrifuged at 4 °C, and worm pellets were lyophilized and extracted with 35 mL of 95% ethanol at room temperature for 12 h. Extracts were dried in vacuo and reconstituted in 200 μL of methanol. For a detailed description of worm rearing and preparation, see Panda et. al.[46].

**Mass spectrometric analysis**. High-resolution LC-MS analysis was performed on a Dionex 3000 UPLC system coupled with a Thermo Q Exactive high-resolution mass spectrometer as described previously[10,46]. Metabolite extracts were separated using a water-acetonitrile gradient using Agilent ZORBOX Eclispse XDB-C18 rapid resolution column (2.1 × 150 mm, particle size 1.8-micron) maintained at 40 °C. Solvent A: 0.1% formic acid in water; Solvent B: 0.1% formic acid in acetonitrile. A/B gradient started at 5% B for 1.5 min after injection and increased linearly to 100% B at 12.5 min, using a flow rate of 0.5 mL/min. Mass spectrometer parameters: spray voltage 2.9 kV; capillary temperature 320 °C; prober heater temperature 300 °C; sheath, auxiliary, and spare gas 70, 2, and 0, respectively; S-

lens RF level S5, resolution 140,000 at *m/z* 200, AGC target $1 \times 10^6$. The instrument was calibrated in negative and positive modes with *m/z* range from 200 to 1000 using calibration solutions (Thermo-Fisher). Ascarosides were detected in negative ionization mode as [M-H]⁻ and MS/MS spectra and retention times confirmed by comparison with known standards.

**MS feature detection and characterization**. LC-MS raw files obtained from at least triplicate sets of plant tissues, unless indicated otherwise, were converted into the mzXML data format (centroid mode) using MSConvert (ProteoWizard), followed by analysis using a customized XCMS R-script based on *matchFilter centWave* algorithm to extract all features[28]. The resulting table of all detected features was used to compare the peak area of ascarosides. Identified ascarosides masses were put on the inclusion list for MS/MS (ddMS2) characterization and checked for the presence of ascaroside diagnostic mass (73.028). Parameters of MS/MS were MS1 resolution 70,000, maximum injection time 250 ms, MS2 resolution 35,000, maximum injection time 125 ms, isolation window 0.8 *m/z*, stepped normalization collision energy 20, 40, 60 or 20, 40, 80, under fill ratio 2.0%, dynamic exclusion 1 s.

**Ascaroside treatments**. Ascarosides (ascr#1[29], ascr#9[29], ascr#10[29], and ascr#18[8]) were dissolved in ethanol to prepare millimolar stock solutions. Final aqueous ascaroside dilutions were prepared fresh on the day of the experiment. Aqueous control solutions contained equal amounts of ethanol (<0.1% for most experiments). For root treatment, plant pots were placed in a tray containing control solution or water supplemented with ascr#18. For seedlings treatment, plant growth media were supplemented with control or ascr#18-containing solutions.

**Soil experiments**. Soil samples were collected from a site (Beebe lake natural area, Ithaca, NY; latitude: 42.4488983154 and longitude: -76.4729003906) free from pesticide and fertilizer application. The collected soil was mixed with potting soil (1:1 mix of field soil: potting mix soil to ensure aeration and nutrition for plant growth) and autoclaved. The soil mix was then used for seeding and plant growth. Three-week-old *Arabidopsis* plants were treated by supplementing soil with mock or ascr#18 solutions (5 mL/pot containing nearly 30–40 *Arabidopsis* plants or 8–10 tomato plants) with three replicates from each experiment. After 2 days, roots of nearly 100 *Arabidopsis* plants per sample and 30 tomato plants per sample were pooled together. Soil was carefully removed from roots by washing with ethanol before sample extraction, as described above. The extracts were dried in vacuo, resuspended in 100–200 μL of methanol and analyzed by LC/MS. The experiment was independently performed three times for *Arabidopsis* (nine biological replicates from three independent experiments) and once for tomato (three biological replicates from one experiment).

**Plant infection assays**. For bacterial growth assays, two leaves of 3.5-week-old *Arabidopsis* plants root pretreated for 24 h with ascr#18 or mock solutions were syringe infiltrated with a suspension of virulent *Pst* DC3000 in 10 mM MgCl₂ at a density of $1 \times 10^5$ colony-forming units (cfu)/mL. For bacterial counts, six leaf discs (diameter of 0.7 cm) from three independently infiltrated plants (two leaf discs/plant) were collected at 3 days post inoculation and placed into a tube. Bacterial recovery was done using 1 mL of 10 mM MgCl₂, serial dilutions, and subsequent plating was performed by the method described previously in Tian et al.[47]. For serial dilutions and plating, 20 μL from each tube was transferred to the wells of a microtitre plate containing 180 μL of 10 mM MgCl₂, and serial tenfold dilutions were prepared using a multi-channel pipette. Five microliter drops from each dilution were spotted onto a 150 mm Petri plate of Luria-Bertani broth (BD Biosciences) media containing 34 μg/mL rifampicin and 15 g/L bacto agar (BD Biosciences), and the plates were incubated at 28 °C. Bacterial counts were performed 48 h post incubation. Three independent repeats were performed at three different times.

For *Arabidopsis* nematode infection assays, *Meloidogyne incognita* (collected in North Carolina, USA[48]) was propagated on tomato (*Solanum lycopersicum* cv. Tiny Tim). *M. incognita* eggs was then isolated from egg masses on tomato roots with 0.5% sodium hypochlorite and rinsed with water on a 25-μm sieve. Collected nematode eggs was treated in a solution of 0.02% sodium azide for 20 min and then hatched over water containing 1.5 mg/mL gentamycin sulfate and 0.05 mg/mL nystatin at room temperature on a Baermann pan for 3 days. Hatched second-stage juveniles (J2) were collected, surface sterilized with an aqueous solution of mercuric chloride (0.004%) and sodium azide (0.004%) for 10 min, and rinsed three times with sterile distilled water. Surface-sterilized J2 were resuspended in 0.1% agarose at a concentration of 10 J2 larvae per 10 μL and used for *Arabidopsis* inoculation. *Arabidopsis* ecotype Col-0 seeds were surface sterilized and planted in six-well plates containing Knop medium supplemented with 2% sucrose. Plants were grown at 24 °C under 16-h-light/8-h-dark conditions. Two microliters of aqueous ≤ 0.1% ethanol solution containing various concentrations of ascr#18 or the aqueous ≤ 0.1% ethanol only control solution was added to each well containing 10-day-old seedlings. After 48 h of pretreatment, the solution was removed and ~300 freshly hatched, surface-sterilized juveniles (J2) of *M. incognita* were inoculated on the roots of each seedling. Galls for *M. incognita* were counted under a microscope 6 weeks after inoculation. Some of the above procedures have been described previously[8]. All experiment were performed with at least three independent repeats at different times.

**Worm attraction assays**. Chemotaxis plates were prepared by pouring 8 mL of 2% bacto-agar (BD Biosciences) into 6-cm Petri dishes. Immediately before adding worms, 10 μL of aqueous ascaroside-containing solutions or control solution were placed on opposite sides of the plate as shown in Fig. 5c. About 100 worms were placed in the center of the plates. The plates were then placed in a 25 °C incubator for 4 h before counting the worms on both sides of the plates. Worms that remained in the center 0.5-cm-wide strip were not counted. The attraction index was calculated as $(A – B/A + B)$, where $A$ and $B$ denote numbers of nematodes on the test solution side and mock solution side, respectively[38]. All experiments were independently performed four times.

**RNA analyses**. For each replicate, total RNA from *Arabidopsis* leaves were isolated from a pool of one leaf from each of three plants with at least two samples from each experiment, and the experiment itself was independently performed at least three times (at least five biological replicates from at least three independent experiments). For the root RNA analyses, *Arabidopsis* roots were collected from 30–40 seedlings for each replicates with at least one sample from each experiment, and the experiment was independently performed three times (at least four biological replicates from three independent experiments). Total RNA was isolated using Qiagen RNeasy Plant Mini Kit (Qiagen) according to the manufacturer's instructions. DNAse treatment was done using DNA-free$^{TM}$ Kit (Ambion) following the manufacturer's instructions. First-strand complementary DNA (cDNA) was synthesized from 1 μg of total RNA using M-MLV reverse transcriptase (Promega) and amplified using gene-specific primers (Supplementary Table 1) and previously listed in Manosalva et al.[8]. For quantitative real-time PCR (qRT-PCR), transcripts were amplified using SYBR Premix Ex-Taq (Takara) from 2.5 μL of 5 × -diluted cDNA in a total 20 μL reaction using 0.1 μL of 100 μM gene-specific primers. Reactions were done using a CFX96 Touch Bio-Rad Real-Time PCR System (Bio-Rad). The PCR conditions were 50 °C for 2 min, 95 °C for 2 min (initial denaturation) followed by 40 cycles of amplification (95 °C for 15 s, 60 °C for 60 s), followed by generation of a dissociation curve. At least, three technical replicates were performed for each of the two or three biological replicates. The transcript levels of defense–response genes in *Arabidopsis* are shown as fold change relative to mock-treated plants. The relative fold change was calculated according to the $2^{-\Delta\Delta Ct}$ method[8,49]. *Ubiquitin10* (AT4G053320) was used as an endogenous reference gene. Two-tailed $t$-tests with an $\alpha$ level of 0.05 were used to compare transcript levels.

**RNA-Seq library construction and sequencing**. For the root RNA sequencing analyses, *Arabidopsis* roots were collected from 30–40 seedlings for each replicate with two samples from each experiment, and the experiment was independently performed three times (total six biological replicates from three independent experiments). The 3′RNA-Seq libraries were prepared from ~500 ng of total RNA per sample using the Lexogen QuantSeq 3′mRNA-Seq Library Prep Kit FWD from Illumina (https://www.lexogen.com/quantseq-3mrna-sequencing/). The libraries were quantified on a plate reader with intercalating dye and pooled for consistency. The pool was then quantified by digital PCR and sequenced on an Illumina NextSeq500 sequencer, single-end 1 × 86 bp, and de-multiplexed based upon six base i7 indices using Illumina bcl2fastq2 software (version 2.17; Illumina, Inc., San Diego, CA).

**Quantitative expression analysis**. Illumina adapters were removed from the de-multiplexed fastq files using Trimmomatic[50] (version 0.36). Poly-A tails and poly-G stretches of at least ten bases in length were then removed using the BBDuk program in the package BBMap [https://sourceforge.net/projects/bbmap/]. The reads were then aligned to the *Arabidopsis thaliana* genome[51] using the STAR aligner[52] (version 2.5.3a). For the STAR indexing step, the Araport11 gff annotation file was converted to gtf format with the gffread program from cufflinks[53] (version 2.2.1). The output SAM files were then converted to BAM using SAM-tools[54] (version 1.6), and the number of reads overlapping each gene in the gff3 file on the forward strand was counted using HTSeq-count[55] (version 0.6.1). Differentially expressed gene (DEG) analysis was performed using the DESeq2 R package 1.24.0[56]. RNA-Seq counts were normalized as Transcripts per million (TPM) and genes with an adjusted $P$-value determined to be <0.05 (false-discover rate < 0.05) a fold change value ≥ 2 (|Log2 fold change| ≥ 1) between two groups were considered to be differentially expressed. The RNA-Seq data is deposited to National Center for Biotechnology information (NCBI) (Sequence Read Archive (SRA) submission: SUB5844685).

**Statistics**. All data are expressed as mean ± SEM. Statistical analysis was conducted between using a student $t$-test, one-way or two-way analysis of variance (ANOVA) with a Tukey post hoc multiple comparisons test, as indicated in the Figure legends. A $p$-value < 0.05 was considered significant. Statistical analyses were performed using either Microsoft Excel (Microsoft, Redmond, WA) or GraphPad Prism, version 8 (GraphPad Software, La Jolla, CA).

**Reporting summary**. Further information on research design is available in the Nature Research Reporting Summary linked to this article.

## Data availability

The data supporting the findings of this study are available within the manuscript and its Supplementary Files. the source data Underlying Figs. 1d, 2b, 2c, 3c, 4b, 4c, 4d, 4e, 4f, 5a, 5b, and 5c are available as a separate Source Data file. RNA-seq data have been deposited to the NCBI Sequence read archive under accession code PRJNA550121.

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

## Acknowledgements

We thank Bonnie Bartel, Rice University for providing *Arabidopsis thaliana* mutants, Alexander Artyukhin and Oishika Panda for assistance with HPLC-MS analyses. This work was supported by the NIH (R01-AT008764 to F.C.S), USDA (AFRI 2011- 68004-30154 to D.F.K), and USDA-ARS to X.W.

## Author contributions

M.M and F.C.S conceived the research. M.M., D.F.K., and F.C.S. designed the research. M.M., F.J.T., S.C., and A.K. performed the research. Y.K.Z. synthesized [13]C-labeled ascr#18. M.M., F.J.T., S.C., V.M.W., X.W, D.F.K., and F.C.S analyzed the data. M.M., D.F.K., and F.C.S wrote the manuscript.

## Competing interests

Authors M.M., D.F.K., and F.C.S. are co-founders of Ascribe Bioscience, a company that develops plant treatments based on small molecules from microbiota. All other authors declare no competing interests.
