## [Peer Review File · Nature Communications]

Reviewers' comments:

Reviewer #1 (Remarks to the Author):

This paper describes shows that a nematode pheromone (ascr#18) is transformed into a set of pheromones with shorter side chains by plants, which subsequently accumulate around the roots. Two acyl-CoA oxidases are identified that are required for this process to occur in planta. The oxidases are not required for ascr#18-induced pathogen resistance, but reduce ascr#18-induced nematode resistance. This effect is attributed to a repellent effect of ascr#18 together with its shorter catabolite ascr#9. The authors conclude that this is a plant defenses, whereby the plant transforms a PAMP into a repellent molecule.

Overall, I found this an interesting and well-prepared manuscript presenting interesting results on the transformation of a nematode pheromone by plants. However, I have a few concerns regarding the biological realism, and, subsequently, the interpretation and relevance of the observed patterns. Further information (and, most likely, further experiments) seem to be required to provide a solid footing for the observed phenomena.

1) Evidence is presented that plants transform ascr#18 when this pheromone is applied in vitro. However, the application of pure ascr#18 into sterile-grown plants may not be a relevant biological situation. I did not find any information on how the doses used correspond to quantities of ascr#18 that are released by biologically relevant densities of nematodes. As the presented phenomenon is new, it would be important to demonstrate that ascr#18 is also metabolized, resulting in the accumulation of other ascarosides, in a realistic setting (i.e. in soil grown plants exposed to natural densities of nematodes).

2) The following experiments should then be conducted using concentrations that are expected to occur in such a biologically relevant setting. This is especially important for the experiments presented in Figs 4 and 5. For instance, is the reduced attraction observed upon application of 10 nM ascr#18 and 100 nM ascr#9 biologically meaningful? If these doses can be justified through actual biologically relevant data, then this information should be provided and discussed in the manuscript. If this data is not available, it needs to be generated, and doses should be chosen accordingly to make this story biologically meaningful.

3) The data generated from using the acx1/acx5 mutant is interesting and relevant. However, without chemical complementation, it remains difficult to come to clear conclusions whether the phenotypes observed in the mutant are indeed the result of a lack of conversion of ascr#18. I would expect this mutant to have an altered hormonal network, and thus to show pleiotropic effects. Indeed, it seems that acx1/5 mutants are defective in wound-induced jasmonate accumulation (<https://www.ncbi.nlm.nih.gov/pubmed/17172287/>). This aspect needs to be addressed in the manuscript. In fact, I do not understand why this fact was simply omitted. In any case, the acx1/acx5 should be complemented with the missing ascarosides to test whether the wild type phenotypes can be recovered. Such experiments are crucial to link the acx1/acx5 to ascr#18 metabolism. Also, demonstrating the activity of the two enzymes towards ascr#18 would be very helpful.

4) Based on the fact that ascr#18-induced defense signaling is not altered in the acx1/acx5 mutant in the leaves, the authors conclude that ascr#18-induced resistance in the roots is unlikely to be due to induced defense signaling. To draw these conclusions, experiments would need to be conducted using root tissues as well as nematode-induction treatments.

Minor comments:

Line numbers are missing in the manuscript. Providing line numbers would greatly aid in providing specific comments.

Abstract: "is employed by" suggests an adaptive context. I don't think the manuscript provides sufficient evidence to warrant this choice of words.

"as a defense mechanism" is again speculative.

Introduction: "Potently" is subjective and should be removed

Last part of introduction: Repeats the abstract instead of providing a rationale for the chosen approach.

Results: Clarify doses and their biological relevance

Leaf-infiltration: I don't see how this is a test regarding endophytes, as leaves may also contain endophytic bacteria in the leaf apoplast?

ACX1/ACX5: Explain primary function and cite relevant literature.

The discussion could benefit from a broader view on the topic, including studies showing how DAMPs are metabolized into resistance factors above ground (e. g.

<https://www.pnas.org/content/111/19/7144>).

Reviewer #2 (Remarks to the Author):

In this manuscript, Manohar and co-authors studied the mode of action of the major ascaroside, ascr#18, secreted by plant-parasitic nematodes. This pheromone has previously been shown to elicit hallmark plant defense responses in leaves including the expression of genes associated with PAMP/MAMP-triggered immunity (Manosalva et al., 2015; DOI: 10.1038/ncomms8795). Using comparative metabolomics of plant tissues and root exudates, they convincingly showed ascr#18 is rapidly metabolized by three plant species into shorter side-chained ascaroside. A screen of fatty acid metabolism mutants in Arabidopsis revealed that a mutant defective in two peroxisomal acyl-CoA oxidases does not metabolize ascr#18. The acx1 acx5 double mutant is defective in ascr#18-mediated defense against nematode, whereas enhanced protection against leaf infection with a bacterial pathogen remains unaffected. Thus, the ascr#8 metabolism is only required for the enhanced resistance of ascr#18-treated plants to nematode infection. Finally, they revealed that shorter side-chained ascarosides are excreted via the roots, which in combination with ascr#18 repel nematodes. Thus, this original work reveals plant editing of nematode pheromones as a defense mechanism. The manuscript is very well written, clear, and convincing. This study concerns a highly interesting topic for plant pathologists and is relevant to Nature Communication scope.

My main concerns are

#1. The authors showed that the activation of defense signaling pathways is independent of ascr#18 metabolism. The data presented are convincing for the leaves but what about the defense response in the roots (target organs of the nematodes)?

#2. I am wondering if the authors have tested other single or double mutants in the ACX genes, and in particular other acx1 acx5 alleles or a complemented line. A single line (acx1-2 acx5-1) is presented. RNAseq experiments of the acx1 acx5 double mutant were carried out. However, data is not presented (except for ACX1 and ACX5) and not deposited in a public repository.

#3. Statistical tests could be performed systematically (appear absent in data presented in Fig 1d; 3c; 5a; 5d; suppl fig 5, 6). I would suggest clarifying in the figure legends the number of plants/samples tested and the number of independent experiments carried out.

Here are additional suggestions

#4. The method section does not indicate how these ascarosides are produced.

#5. I am questioning whether the Fig 1a and 1b are required. Fig 1a appears redundant with fig 6, and the structures presented in 1b are presented in fig 1c.

Reviewer #3 (Remarks to the Author):

In this manuscript, the authors find that the nematode pheromone ascr#18 is chain-shortened in plant roots (Figure 1, 2) and leaves (Figure 2) via peroxisomal beta-oxidation (Figure 3). In Arabidopsis, this conversion requires the peroxisomal acyl-CoA oxidase isozymes ACX1 and/or ACX5, but not the enoyl-CoA hydratase-like enzymes ECH2 or IBR10 (Figure 3). Additional

experiments with the *acx1 acx5* double mutant imply that *ascr#18* chain shortening by the plant is needed for applied *acr#18* to deter nematodes from roots (Figure 4d, e) and prevent gall formation (Figure 4c) but not to dampen growth of a bacterial pathogen (Figure 4b) or induce defense-response genes (Figure 4f). Finally, the authors show that certain combinations of *ascr#18* and its chain-shortened derivative *ascr#9* can repel nematodes even in the absence of plant roots (Figure 5). The authors conclude that the plant is editing a pathogen-derived chemical messenger to change the message, which is a fascinating finding.

Major points:

1. p. 6 - The reader would benefit from being told the rationale for immediately testing *acx1 acx5* rather than single *acx1* or *acx5* mutants or other *acx* mutants.
2. The *acx1 acx5* double mutant is deficient in jasmonate synthesis in response to wounding or herbivore attack and is less resistant to multiple arthropod pests (e.g., Schilmiller et al., 2007, *Plant Physiol* 143: 812). This paper is not discussed or cited. It would seem that a deficiency of JA in the *acx1 acx5* mutant could be of some relevance. For example, the muted *ascr#18* induction of some defense response genes in *acx1 acx5* might stem from a failure of a JA-biosynthesis fueled positive feedback loop in the mutant.
3. The *acx1 acx5* double mutant presents significantly fewer root galls following nematode infection than wild type (Figure 4c). This counterintuitive resistance is not discussed in the text. Is JA synthesis required for gall formation (see point 2)? Is there some other hypothesis to explain this result?
4. For (at least) some of the data, it would be preferable to present the actual data points rather than a histogram. In particular, the qRT-PCR data in figure 4f would benefit from this change in presentation, as the s.d. error bars are very large on some of the points. The methods indicate that the qRT-PCR used three technical replicates of two or three biological replicates (p. 15). The legend to Figure 4f indicates that $n \geq 6$. Is this n for a mix of biological and technical replicates? If so, it would seem that the individual data points should be shown to avoid combining biological and technical replicates; biological replicates could be different colors, for example.

Minor points:

5. It may be that ANOVA would be more appropriate than t-tests for many of the experiments comparing more than two conditions.
6. Given that *ascr#1* and *ascr#9* are the most abundant plant-produced derivatives of *ascr#18* (Figure 5a), some rationale for using *ascr#9* in combination with *ascr#18* rather than with *ascr#1* in Figure 5d would be useful to the reader.
7. p. 2 - The use of "microbiota" in the last sentence of the abstract is going to suggest bacteria and fungi to many readers. "Organisms" would more clearly include nematodes, which is the authors' intent.
8. p. 6 - The sentence "In contrast, two other putative β o genes, *ibr10* and *ech2*, a putative enoyl-CoA hydratases, are not required for *ascr#18* metabolism in *Arabidopsis*" would be more clear as "In contrast, two other β o enzymes that resemble enoyl-CoA hydratases, *IBR10* and *ECH2*, are not required for *ascr#18* metabolism in *Arabidopsis*."
9. The RNA-seq experiment is insufficiently described. The Y-axis of supplemental Figure 5 is labeled "transcript abundance." Presumably these data are normalized relative to something, which should be specified in the legend or axis label.

10. p. 7 – “jasmonic acid (SA)” should be “jasmonic acid (JA)”.
11. “Ubiquitin was used as an endogenous reference gene” (p. 15) for the qRT-PCR. This is an insufficient description; there are many *Arabidopsis* genes that encode ubiquitin.
12. Figure 6 – It seems odd to introduce the term “semiotic defense” in the figure without using the term in the text.
13. The authors abbreviate “peroxisomal β -oxidation” as “p β o”, and define the abbreviation three times in the text (p. 4, 5, 10). Most authors do not abbreviate this term, and more abbreviations make the manuscript less accessible.

Response to Reviewers' comments:

Reviewer #1 (Remarks to the Author):

This paper describes shows that a nematode pheromone (ascr#18) is transformed into a set of pheromones with shorter side chains by plants, which subsequently accumulate around the roots. Two acyl-CoA oxidases are identified that are required for this process to occur in planta. The oxidases are not required for ascr#18-induced pathogen resistance, but reduce ascr#18-induced nematode resistance. This effect is attributed to a repellent effect of ascr#18 together with its shorter catabolite ascr#9. The authors conclude that this is a plant defenses, whereby the plant transforms a PAMP into a repellent molecule.

Overall, I found this an interesting and well-prepared manuscript presenting interesting results on the transformation of a nematode pheromone by plants. However, I have a few concerns regarding the biological realism, and, subsequently, the interpretation and relevance of the observed patterns. Further information (and, most likely, further experiments) seem to be required to provide a solid footing for the observed phenomena.

1) Evidence is presented that plants transform ascr#18 when this pheromone is applied in vitro. However, the application of pure ascr#18 into sterile-grown plants may not be a relevant biological situation. I did not find any information on how the doses used correspond to quantities of ascr#18 that are released by biologically relevant densities of nematodes. As the presented phenomenon is new, it would be important to demonstrate that ascr#18 is also metabolized, resulting in the accumulation of other ascarosides, in a realistic setting (i.e. in soil grown plants exposed to natural densities of nematodes).

Response: We have previously demonstrated that the concentration of ascr#18 in the *Meloidogyne* spp. culture media is between 5 nM to 100 nM (Manosalva et al., 2015). In addition, we have now shown that the metabolism of ascr#18 to ascr#9 in a field soil-potting soil mixture is comparable to that in sterile growth media, and that root exudates from plants infested with nematodes contain both ascr#9 and ascr#18, in a ratio similar to that measured when treating plants with synthetic ascr#18 (Supplementary Fig. 4).

2) The following experiments should then be conducted using concentrations that are expected to occur in such a biologically relevant setting. This is especially important for the experiments presented in Figs 4 and 5. For instance, is the reduced attraction observed upon application of 10 nM ascr#18 and 100 nM ascr#9 biologically meaningful? If these doses can be justified through actual biologically relevant data, then this information should be provided and discussed in the manuscript. If this data is not available, it needs to be generated, and doses should be chosen accordingly to make this story biologically meaningful.

Response: As reported (Manosalva et al., 2015), concentrations of ascr#18 accumulating in growth media after 24 h at a density of 5-10,000 worms per mL are in the low nanomolar range, and these are the concentrations used in the assays. Actual concentrations of ascarosides in the rhizosphere are bound to be highly variable. In the rhizosphere of a nematode-infected plant, ascaroside concentrations will follow a gradient from highest in densely infected root mass to near zero in the periphery. Densely infected root mass contains more than 1000 nematodes per gram of soil, which assuming a liquid content of 20% closely corresponds to the densities in the growth media assays.

3) The data generated from using the acx1/acx5 mutant is interesting and relevant. However, without chemical complementation, it remains difficult to come to clear conclusions whether the phenotypes observed in the mutant are indeed the result of a lack of conversion of ascr#18. I would expect this mutant

to have an altered hormonal network, and thus to show pleiotropic effects. Indeed, it seems that acx1/5 mutants are defective in wound-induced jasmonate accumulation (<https://www.ncbi.nlm.nih.gov/pubmed/17172287/>). This aspect needs to be addressed in the manuscript. In fact, I do not understand why this fact was simply omitted. In any case, the acx1/acx5 should be complemented with the missing ascarosides to test whether the wild type phenotypes can be recovered. Such experiments are crucial to link the acx1/acx5 to ascr#18 metabolization. Also, demonstrating the activity of the two enzymes towards ascr#18 would be very helpful.

Response: In this revision we added qRT-PCR and RNA-seq data showing that defense-associated hormonal pathways including JA, SA, auxin, and ethylene are unaffected by low (50 nM) ascr#18 concentrations in wildtype and remained largely unchanged in the acx1acx5 double mutant (Figure 4f and Supplementary Fig. 8). These data argue that it is unlikely that the phenotype observed in acx1acx5 is due to pleiotropism.

Adding ascr#18/ascr#9 mixes back to acx1/acx5 seems unlikely to recover the WT phenotype. Our data (Figure 5) show that nematodes migrate away from this combination, along the concentration gradient, which would be absent in potted acx1/acx5 plants treated with ascr#18/ascr#9.

4) Based on the fact that ascr#18-induced defense signaling is not altered in the acx1/acx5 mutant in the leaves, the authors conclude that ascr#18-induced resistance in the roots is unlikely to be due to induced defense signaling. To draw these conclusions, experiments would need to be conducted using root tissues as well as nematode-induction treatments.

Response: We have now conducted expression analysis in roots (please refer to Supplementary Figure 8).

Minor comments:

Line numbers are missing in the manuscript. Providing line numbers would greatly aid in providing specific comments.

Response: Line numbers have been added.

Abstract: “is employed by” suggests an adaptive context. I don’t think the manuscript provides sufficient evidence to warrant this choice of words.

Response: “is employed by” has been replaced with “is metabolized by”.

“as a defense mechanism” is again speculative.

Response: We revised this sentence to state that our data suggest that ascaroside metabolism “serves as a defense mechanism”.

Introduction: “Potently” is subjective and should be removed

Response: “Potently” has been removed.

Last part of introduction: Repeats the abstract instead of providing a rationale for the chosen approach.

Results: Clarify doses and their biological relevance

Response: We removed repetition of information already presented in the Abstract from the last part of the introduction and clarified choice of concentrations throughout the text.

Leaf-infiltration: I don’t see how this is a test regarding endophytes, as leaves may also contain endophytic bacteria in the leaf apoplast?

Response: We agree. The leaf assays merely exclude the root-associated microorganisms as likely participants. This section has been revised accordingly..

ACX1/ACX5: Explain primary function and cite relevant literature.

Response: We added information on the roles of ACX1 and ACX5 and cited relevant literature.

The discussion could benefit from a broader view on the topic, including studies showing how DAMPs are metabolized into resistance factors above ground (e. g. <https://www.pnas.org/content/111/19/7144>).

Response: We have now included this reference as a second example, together with that of flagellin deglycosylation, which suggest that plants may utilize a broad spectrum of metabolic transformations to edit foreign signals for their protection.

Reviewer #2 (Remarks to the Author):

In this manuscript, Manohar and co-authors studied the mode of action of the major ascaroside, ascr#18, secreted by plant-parasitic nematodes. This pheromone has previously been shown to elicit hallmark plant defense responses in leaves including the expression of genes associated with PAMP/MAMP-triggered immunity (Manosalva et al., 2015; DOI: 10.1038/ncomms8795). Using comparative metabolomics of plant tissues and root exudates, they convincingly showed ascr#18 is rapidly metabolized by three plant species into shorter side-chained ascaroside. A screen of fatty acid metabolism mutants in Arabidopsis revealed that a mutant defective in two peroxisomal acyl-CoA oxidases does not metabolize ascr#18. The acx1 acx5 double mutant is defective in ascr#18-mediated defense against nematode, whereas enhanced protection against leaf infection with a bacterial pathogen remains unaffected. Thus, the ascr#8 metabolism is only required for the enhanced resistance of ascr#18-treated plants to nematode infection. Finally, they revealed that shorter side-chained ascarosides are excreted via the roots, which in combination with ascr#18 repel nematodes. Thus, this original work reveal plant editing of nematode pheromones as a defense mechanism. The manuscript is very well written, clear, and convincing. This study concerns a highly interesting topic for plant pathologists and is relevant to Nature Communication scope.

My main concerns are

#1. The authors showed that the activation of defense signaling pathways is independent of ascr#18 metabolism. The data presented are convincing for the leaves but what about the defense response in the roots (target organs of the nematodes)?

Response: We have now performed gene expression analyses in roots – please see our response to reviewer 1 above and Supplementary Fig. 8. In addition, we have deposited the root RNA sequencing dataset at NCBI.

#2. I am wondering if the authors have tested other single or double mutants in the ACX genes, and in particular other acx1 acx5 alleles or a complemented line. A single line (acx1-2 acx5-1) is presented. RNAseq experiments of the acx1 acx5 double mutant were carried out. However, data is not presented (except for ACX1 and ACX5) and not deposited in a public repository.

Response: To address this concern, we have performed additional experiments with single mutants and found that both acx1 and acx5 are capable of metabolizing ascr#18 into ascr#9, similar to wildtype (Supplementary Fig. 7).

#3. *Statistical tests could be performed systematically (appear absent in data presented in Fig 1d; 3c; 5a; 5d; suppl fig 5, 6). I would suggest clarifying in the figure legends the number of plants/samples tested and the number of independent experiments carried out.*

Response: Thank you for the suggestion. We have now added data points in all figures and added p values where indicated.

Here are additional suggestions

#4. *The method section does not indicate how these ascarosides are produced.*

Response: ascr#18 was synthesized as described before (Manosalva et al., 2015). The reference is now included in the Methods section.

#5. *I am questioning whether the Fig 1a and 1b are required. Fig 1a appears redundant with fig 6, and the structures presented in 1b are presented in fig 1c.*

Response: We have included Figure 1a to present current knowledge of pattern-triggered immunity in plants (helpful for non-plant science readers). The structures shown in Figure 1b (not the same as in 1c) are included to illustrate structural diversity among nematode ascarosides.

Reviewer #3 (Remarks to the Author):

In this manuscript, the authors find that the nematode pheromone ascr#18 is chain-shortened in plant roots (Figure 1, 2) and leaves (Figure 2) via peroxisomal beta-oxidation (Figure 3). In Arabidopsis, this conversion requires the peroxisomal acyl-CoA oxidase isozymes ACX1 and/or ACX5, but not the enoyl-CoA hydratase-like enzymes ECH2 or IBR10 (Figure 3). Additional experiments with the acx1 acx5 double mutant imply that ascr#18 chain shortening by the plant is needed for applied acr#18 to deter nematodes from roots (Figure 4d, e) and prevent gall formation (Figure 4c) but not to dampen growth of a bacterial pathogen (Figure 4b) or induce defense-response genes (Figure 4f). Finally, the authors show that certain combinations of ascr#18 and its chain-shortened derivative ascr#9 can repel nematodes even in the absence of plant roots (Figure 5). The authors conclude that the plant is editing a pathogen-derived chemical messenger to change the message, which is a fascinating finding.

Major points:

1. p. 6 - The reader would benefit from being told the rationale for immediately testing acx1 acx5 rather than single acx1 or acx5 mutants or other acx mutants.

Response: The acx1acx5 double mutant was tested first since previous work indicated that these two acx function redundantly. In this revision we added data for the single mutants showing that ascr#18 metabolism was largely unimpaired, confirming redundancy (see Supplementary Fig.7).

2. The acx1 acx5 double mutant is deficient in jasmonate synthesis in response to wounding or herbivore attack and is less resistant to multiple arthropod pests (e.g., Schilmiller et al., 2007, Plant Physiol. 143: 812). This paper is not discussed or cited. It would seem that a deficiency of JA in the acx1 acx5 mutant could be of some relevance. For example, the muted ascr#18 induction of some defense response genes in acx1 acx5 might stem from a failure of a JA-biosynthesis fueled positive feedback loop in the mutant.

Response: Thank you for this suggestion. In this revision, we included a discussion of Schillmiller et al., 2007 and added root gene expression results indicating that plant defense signaling pathways, including JA, are not significantly altered in *acx1acx5*.

3. *The acx1 acx5 double mutant presents significantly fewer root galls following nematode infection than wild type (Figure 4c). This counterintuitive resistance is not discussed in the text. Is JA synthesis required for gall formation (see point 2)? Is there some other hypothesis to explain this result?*

Response: There is a slight reduction in the number of galls; however, this effect is small compared to the effects of ascarosides. Although minor perturbation of e.g. JA or auxin signaling cannot be excluded as a cause, we did not discuss this in the text, given that the effect is borderline significant and any hypotheses about causation would have to remain highly speculative.

4. *For (at least) some of the data, it would be preferable to present the actual data points rather than a histogram. In particular, the qRT-PCR data in figure 4f would benefit from this change in presentation, as the s.d. error bars are very large on some of the points. The methods indicate that the qRT-PCR used three technical replicates of two or three biological replicates (p. 15). The legend to Figure 4f indicates that $n \geq 6$. Is this n for a mix of biological and technical replicates? If so, it would seem that the individual data points should be shown to avoid combining biological and technical replicates; biological replicates could be different colors, for example.*

Response: We have changed the Figures to indicate number of independent samples used in the analyses. The number of biological replicates and number of independent experiments are described in the method section under RNA analyses.

Minor points:

5. *It may be that ANOVA would be more appropriate than t-tests for many of the experiments comparing more than two conditions.*

Response: The reviewer is correct. We now use ANOVA for all multiple comparisons.

6. *Given that ascr#1 and ascr#9 are the most abundant plant-produced derivatives of ascr#18 (Figure 5a), some rationale for using ascr#9 in combination with ascr#18 rather than with ascr#1 in Figure 5d would be useful to the reader.*

Response: For bioassays with nematodes, we chose a combination of ascr#18, which is directly excreted by nematodes, and the most abundant plant metabolite, ascr#9. It is possible that intermediates in the pathway from ascr#18 to ascr#9 also contribute, e.g. ascr#1, which is excreted alongside ascr#9 in the root exudates, and this is now mentioned in the Discussion. We also revised the wording in the Results section to make the rationale for our choice clearer.

7. *p. 2 - The use of “microbiota” in the last sentence of the abstract is going to suggest bacteria and fungi to many readers. “Organisms” would more clearly include nematodes, which is the authors’ intent.*

Response: We agree and have replaced “microbiota” with “organisms”.

8. *p. 6 - The sentence “In contrast, two other putative p β o genes, *ibr10* and *ech2*, a putative enoyl-CoA hydratases, are not required for ascr#18 metabolism in *Arabidopsis*” would be more clear as “In contrast, two other p β o enzymes that resemble enoyl-CoA hydratases, *IBR10* and *ECH2*, are not required for ascr#18 metabolism in *Arabidopsis*.”*

Response: Thank you for the suggestion. We have changed the sentence as suggested.

9. *The RNA-seq experiment is insufficiently described. The Y-axis of supplemental Figure 5 is labeled “transcript abundance.” Presumably these data are normalized relative to something, which should be specified in the legend or axis label.*

Response: Methods for analysis of the RNA-seq data are now described in Methods, and the dataset has been deposited in NCBI public repository.

10. *p. 7 – “jasmonic acid (SA)” should be “jasmonic acid (JA)”.*

Response: Corrected, thanks.

11. *“Ubiquitin was used as an endogenous reference gene” (p. 15) for the qRT-PCR. This is an insufficient description; there are many Arabidopsis genes that encode ubiquitin.*

Response: We used *Ubiquitin 10 (AT4G05320)* as the reference gene; this is now clarified in the Methods section.

12. *Figure 6 – It seems odd to introduce the term “semiotic defense” in the figure without using the term in the text.*

Response: We have incorporated “semiotic” in the text as suggested.

13. *The authors abbreviate “peroxisomal β -oxidation” as “ $p\beta o$ ”, and define the abbreviation three times in the text (p. 4, 5, 10). Most authors do not abbreviate this term, and more abbreviations make the manuscript less accessible.*

Response: We have removed the abbreviation “ $p\beta o$ ” as suggested.

REVIEWERS' COMMENTS:

Reviewer #1 (Remarks to the Author):

The authors have answered my questions, and I am, overall, satisfied with the revisions.

One aspect that should be improved is the statistics used to test the impact of the *acx1acx5* mutation on *ascr#18* induced defense expression. The authors use simple t-tests, when the appropriate test would be Two-Way ANOVAs testing for interactions between treatment and genotype. Same thing for Fig. 4b.

Minor comment:

Line 24: Remove "thus".

Reviewer #2 (Remarks to the Author):

This article is a resubmission of an article I evaluated for Nature Comm in March 2019. As I previously wrote, in this manuscript, Manohar and co-authors studied the mode of action of the major ascaroside, *ascr#18*, secreted by plant-parasitic nematodes. This pheromone has previously been shown to elicit hallmark plant defense responses in leaves including the expression of genes associated with PAMP/MAMP-triggered immunity (Manosalva et al., 2015; DOI: 10.1038/ncomms8795). Using comparative metabolomics of plant tissues and root exudates, they convincingly showed *ascr#18* is rapidly metabolized by three plant species into shorter side-chained ascaroside. A screen of fatty acid metabolism mutants in *Arabidopsis* revealed that a mutant defective in two peroxisomal acyl-CoA oxidases does not metabolize *ascr#18*. The *acx1 acx5* double mutant is defective in *ascr#18*-mediated defense against nematode, whereas enhanced protection against leaf infection with a bacterial pathogen remains unaffected. Thus, the *ascr#8* metabolism is only required for the enhanced resistance of *ascr#18*-treated plants to nematode infection. Finally, they revealed that shorter side-chained ascarosides are excreted via the roots, which in combination with *ascr#18* repel nematodes. Thus, this original work reveals plant editing of nematode pheromones as a defense mechanism. The manuscript is very well written, clear, and convincing. This study concerns a highly interesting topic for plant pathologists and is relevant to Nature Communication scope.

I thank the authors for taking into consideration all my remarks. The new elements brought forward are convincing.

Some last minor remarks:

Fig1a Pheromones instead of Phermones?

L406. What is the origin / strain of *M. incognita* used?

Reviewer #3 (Remarks to the Author):

This is a fascinating study demonstrating that plants are editing a pathogen-derived chemical messenger to change the message. The authors have strengthened the manuscript and have addressed my previous concerns. I appreciate seeing the raw data points overlaid on the graphs.

Minor points:

Line 161-162: The phrase "...indicating that metabolism of *ascr#18* via *acx1acx5* is not required

for enhanced resistance against this bacterial pathogen" should be "...indicating that metabolism of ascr#18 via ACX1 or ACX5 is not required for enhanced resistance against this bacterial pathogen."

Throughout: I believe that the convention for double mutants in Arabidopsis is to have a space between the two alleles, thus "acx1acx5" should be "acx1 acx5".

Response to Reviewers' comments:

Reviewer #1 (Remarks to the Author):

The authors have answered my questions, and I am, overall, satisfied with the revisions.

*One aspect that should be improved is the statistics used to test the impact of the *acx1acx5* mutation on *ascr#18* induced defense expression. The authors use simple t-tests, when the appropriate test would be Two-Way ANOVAs testing for interactions between treatment and genotype. Same thing for Fig. 4b.*

Response: The reviewer is correct. We now use two-way ANOVA for all multiple comparisons.

Minor comment:

Line 24: Remove "thus".

Response: Done.

Reviewer #2 (Remarks to the Author):

*This article is a resubmission of an article I evaluated for Nature Comm in March 2019. As I previously wrote, in this manuscript, Manohar and co-authors studied the mode of action of the major ascaroside, *ascr#18*, secreted by plant-parasitic nematodes. This pheromone has previously been shown to elicit hallmark plant defense responses in leaves including the expression of genes associated with PAMP/MAMP-triggered immunity (Manosalva et al., 2015; DOI: 10.1038/ncomms8795). Using comparative metabolomics of plant tissues and root exudates, they convincingly showed *ascr#18* is rapidly metabolized by three plant species into shorter side-chained ascaroside. A screen of fatty acid metabolism mutants in *Arabidopsis* revealed that a mutant defective in two peroxisomal acyl-CoA oxidases does not metabolize *ascr#18*. The *acx1 acx5* double mutant is defective in *ascr#18*-mediated defense against nematode, whereas enhanced protection against leaf infection with a bacterial pathogen remains unaffected. Thus, the *ascr#8* metabolism is only required for the enhanced resistance of *ascr#18*-treated plants to nematode infection. Finally, they revealed that shorter side-chained ascarosides are excreted via the roots, which in combination with *ascr#18* repel nematodes. Thus, this original work reveal plant editing of nematode pheromones as a defense mechanism. The manuscript is very well written, clear, and convincing. This study concerns a highly interesting topic for plant pathologists and is relevant to Nature Communication scope.*

I thank the authors for taking into consideration all my remarks. The new elements brought forward are convincing.

Some last minor remarks:

Fig1a Pheromones instead of Phermones?

Response: Done, thanks!

L406. What is the origin / strain of *M. incognita* used?

Response: We added a reference for the origin of the *M. incognita* strain used in this study (reference 48).

Reviewer #3 (Remarks to the Author):

This is a fascinating study demonstrating that plants are editing a pathogen-derived chemical messenger to change the message. The authors have strengthened the manuscript and have addressed my previous concerns. I appreciate seeing the raw data points overlaid on the graphs.

Minor points:

Line 161-162: The phrase "...indicating that metabolism of ascr#18 via acx1acx5 is not required for enhanced resistance against this bacterial pathogen" should be "...indicating that metabolism of ascr#18 via ACX1 or ACX5 is not required for enhanced resistance against this bacterial pathogen."

Response: Thank you for the suggestion. We have modified the sentence.

Throughout: I believe that the convention for double mutants in Arabidopsis is to have a space between the two alleles, thus "acx1acx5" should be "acx1 acx5".

Response: The reviewer is correct. Fixed.